# The Role of Endocan Expression in the Diagnosis and Grading of Precancerous Gastric Lesions

**DOI:** 10.3390/diagnostics15111379

**Published:** 2025-05-29

**Authors:** Mehmet Kok, Suleyman Dolu, Mehmet Emin Arayici, Gokhan Koker, Kadir Balaban, Ayhan Hilmi Cekin

**Affiliations:** 1Department of Internal Medicine, University of Health Sciences, Antalya Training and Research Hospital, Antalya 07100, Turkey; dr.mehmetkok@hotmail.com (M.K.); gkhnkkr@hotmail.com (G.K.); 2Department of Gastroenterology, Faculty of Medicine, Dokuz Eylül University, İzmir 35340, Turkey; 3Department of Biostatistics and Medical Informatics, Faculty of Medicine, Dokuz Eylül University, Izmir 35340, Turkey; mehmet.e.arayici@gmail.com; 4Department of Pathology, University of Health Sciences, Antalya Training and Research Hospital, Antalya 07100, Turkey; kadirbalaban1453@gmail.com; 5Department of Gastroenterology, University of Health Sciences, Antalya Training and Research Hospital, Antalya 07100, Turkey; ayhancekin@hotmail.com

**Keywords:** endocan, gastric cancer, precancerous gastric lesions, gastric dysplasia

## Abstract

**Background/Aims:** Gastric dysplasia is a critical precursor to gastric cancer (GC), and accurate diagnosis and grading of these lesions are essential for effective surveillance and intervention. However, current diagnostic methods such as forceps biopsy have notable limitations, underscoring the need for reliable biomarkers. This study aimed to evaluate the diagnostic and grading utility of endocan, a soluble proteoglycan secreted by activated endothelial cells, in gastric dysplastic lesions. **Methods:** A total of 72 patients with gastric dysplasia, 80 with gastric adenocarcinoma, and 55 healthy controls were prospectively enrolled. Endocan expression in gastric tissue samples was assessed via immunohistochemistry and semi-quantitatively graded. Statistical comparisons were made between control, dysplastic (low-grade and high-grade), and malignant groups. **Results:** Endocan was negatively expressed in all control subjects and positively expressed in 65.3% of the dysplasia group and 100% of the gastric cancer group (*p* < 0.001). Notably, all high-grade dysplasia cases were endocan-positive, whereas 75.8% of low-grade dysplasia cases were endocan-negative (*p* < 0.001). **Conclusions:** This is the first study to demonstrate that endocan is overexpressed in gastric dysplastic lesions. Tissue endocan expression may serve as a practical and robust marker for the diagnosis and grading of gastric dysplasia, potentially enhancing early detection and risk stratification in gastric carcinogenesis.

## 1. Introduction

Gastric cancer (GC) remains a significant global health burden, ranking as the fifth most commonly diagnosed malignancy and the third leading cause of cancer-related mortality worldwide [1,2]. One of the major challenges in managing GC lies in its frequently late-stage diagnosis, which contributes to its poor overall prognosis. Patients diagnosed with advanced-stage disease typically face a 5-year survival rate of less than 25%, underscoring the limited effectiveness of current therapeutic strategies at later stages. In stark contrast, when gastric cancer is detected at an early stage—often incidentally or through screening programs in high-incidence regions—the 5-year survival rate dramatically increases to 90% or higher, reflecting the crucial importance of early detection and intervention [1].

GC is a multifactorial disease influenced by a complex interplay of environmental, lifestyle, and biological factors, including several nonmodifiable risks such as advancing age, male sex, racial and ethnic background, and genetic predisposition, and it may sometimes present with clinical and laboratory differences depending on its anatomical localization [3,4,5,6,7]. Its development typically follows a well-defined, stepwise histopathological sequence known as the Correa cascade [8]. This multistep process begins with chronic gastritis, characterized by persistent inflammation of the gastric mucosa, and gradually progresses through mucosal atrophy (loss of gastric glandular structures), intestinal metaplasia (replacement of gastric epithelium with intestinal-type epithelium), and dysplasia (neoplastic changes confined to the epithelium), culminating in invasive carcinoma [9]. Each step along this cascade provides a potential window of opportunity for intervention and surveillance. Notably, dysplasia—particularly in its high-grade form—represents a critical juncture, as it significantly increases the risk of malignant transformation. Hence, precise detection, classification, and monitoring of precancerous gastric lesions have become vital components of contemporary gastric cancer prevention strategies.

The World Health Organization (WHO) characterizes dysplasia in the gastrointestinal system as the presence of neoplastic epithelium at the histological level, lacking any signs of tissue invasion. Dysplasia is commonly divided into two categories: low-grade dysplasia (LGD) and high-grade dysplasia (HGD) [9]. The transformation into malignancy within high-grade dysplasia (HGD) has been documented with a reported incidence spanning from 60% to 85% over a median period of 4 to 48 months [10,11,12]. In comparison to HGD, low-grade dysplasia (LGD) is linked to a reduced risk of progressing to carcinoma. Within an average period of 10 to 48 months, the proportion of LGD cases undergoing malignant transformation ranges from 0% to 23% [10]. Today, a forceps biopsy is used to diagnose dysplasia. Nonetheless, a meta-analysis has indicated that as many as 25% of gastric low-grade dysplasias (LGDs) confirmed through forceps biopsy may have been incorrectly diagnosed, potentially representing high-grade dysplasia (HGD) or even gastric carcinoma [13]. Therefore, new biomarkers are needed for a more accurate diagnosis. In this respect, endocan may be used as a new marker in the diagnosis of dysplasia.

Endocan is a soluble proteoglycan produced and released by activated vascular endothelial cells, including those present in tumor tissues [14,15]. In experimental research, endocan has been revealed to stimulate tumor growth and is strongly connected to the conversion of dormant tumors into rapidly expanding angiogenic tumors [16].

Recent studies have shown that endocan is overexpressed in various tumor types, including gastric cancer, colorectal cancer, glioblastoma, pituitary adenoma, nonsmall cell lung cancer, and renal cell cancer. Most studies also suggested that endocan overexpression was associated with aggressive tumor progression and poor outcomes [17]. For this reason, endocan has been identified as a potential novel endothelial cell marker and a new target for cancer therapy [18]. Several studies have previously reported the overexpression of ESM1 in gastric cancer and its potential role as a prognostic marker, associated with tumor stage, lymph node metastasis, and poor clinical outcomes [19,20,21]. However, the potential role of tissue ESM1 expression as a diagnostic and grading marker for precancerous gastric dysplastic lesions remains to be clarified.

This study aimed to investigate whether endocan expression could be used as a potential marker for diagnosing and grading precancerous gastric lesions (dysplasia).

## 2. Material and Methods

### 2.1. Patients

This was a prospective, observational diagnostic accuracy study conducted in accordance with the STARD and STROBE guidelines. Patients were consecutively recruited during routine upper gastrointestinal endoscopy procedures at the University of Health Sciences, Antalya Training and Research Hospital between March 2019 and April 2021. Inclusion in the study was based on histopathological confirmation of gastric dysplasia (low- or high-grade), gastric adenocarcinoma, or normal gastric mucosa. Only individuals who met the eligibility criteria and provided written informed consent were enrolled.

In this study, 72 patients with gastric precancerous lesions, 80 with gastric adenocarcinomas, and 55 healthy participants were enrolled at the University of Health Sciences, Antalya Training and Research Hospital. Five biopsies, corresponding to the updated Sydney system, were obtained. Gastric precancerous and cancerous lesions were histopathologically confirmed, and precancerous lesions were defined as low- and high-grade dysplasia groups. In the precancerous group, 64 gastric tissue biopsies, 38 with low-grade and 26 with high-grade dysplasia, were analyzed. Low-grade and high-grade classification was evaluated according to WHO criteria. The TNM staging system (TNM 8th edition by the American Joint Committee on Cancer) was used for pathological staging [22]. Additionally, 47 individuals without any history of gastric malignancy and displaying no abnormalities during esophagogastroduodenoscopy (EGD) (Olympus Medical Systems Corporation, Tokyo, Japan) and on its associated biopsy were recruited from the Endoscopy Center at the University of Health Sciences, Antalya Training, and Research Hospital to constitute the control group. The exclusion criteria encompassed (1) patients diagnosed with stage T4 gastric cancer due to the potential for deep transmural invasion and adjacent organ involvement, which may interfere with consistent immunohistochemical staining evaluation, (2) prior history of gastric surgery, (3) suspected or concurrent other cancers, (4) previous chemotherapy or radiotherapy, (5) usage of medications known to potentially affect endocan levels, such as angiotensin receptor blockers, calcium channel blockers, statins, and non-steroidal anti-inflammatory drugs, (6) individuals with coexisting conditions that may influence endocan expressions, such as chronic hypertension, renal disorders, and diabetes mellitus, and (7) individuals who were pregnant or lactating.

### 2.2. Study Outcomes

The primary outcome of this study was an evaluation of the diagnostic accuracy of tissue endocan expression in distinguishing gastric dysplastic lesions from normal gastric mucosa and gastric adenocarcinoma, including its ability to differentiate low-grade from high-grade dysplasia. The secondary outcomes included the assessment of endocan expression in relation to gastric cancer staging and the correlation of expression levels with clinicopathological parameters.

### 2.3. Immunohistochemistry

Histological images were captured using an Olympus BX53 microscope (Olympus Corporation, Tokyo, Japan) equipped with a DP74 digital camera. A 40× objective lens (NA 0.75) was used in combination with a 10× ocular lens (total magnification 400×). Tissue sections, each measuring 5 μM in thickness, were prepared from paraffin-embedded blocks, with each paraffin block being dedicated for evaluation. Immunohistochemical staining of endocan was performed using an anti-endocan rabbit monoclonal antibody (ab224591, Abcam, Cambridge, MA, USA) and the Streptavidin peroxidase complex procedure. The slides were deparaffinized with xylene and then washed in Phosphate buffered saline. Following antigen retrieval, the slides were subjected to a 40 min incubation at 37 °C with a 0.3% blocking serum to minimize non-specific binding. The primary antibody against endocan was applied overnight at 4 °C on the tissue sections (dilution: 1:100). After rinsing the tissue sections in Tris-buffered saline (TBS), a secondary antibody, biotinylated goat anti-mouse immunoglobulin, was used for detection. The sections were incubated in a 3,3′-diaminobenzidine solution until the desired staining was achieved. Subsequently, the slides were dehydrated, cleared, and mounted after counterstaining with hematoxylin. Negative control samples were established by omitting the primary antibodies.

### 2.4. Evaluation of Immunohistochemical Stain

For the assessment of endocan as immuno-positive staining, positive cells exhibited a brown color in either the nucleus or cytoplasm. Endocan expression was semi-quantitatively categorized based on the subsequent criteria: considered negative (−) when fewer than 1% of cells distinctly displayed endocan and categorized as positive (+) when at least 1 morphologically unequivocal cell distinctly expressed endocan. The immunohistochemical results for endocan also were classified according to the number of positive cells as follows: (−) if f < 1% of cells were stained, (+) < 25% of cells were stained, (++) 25–50% of cells were stained, and (+++) > 50% of cells were stained. The staining was interpreted by two pathologists independently (Figure 1).

### 2.5. Statistical Analysis

Descriptive statistics were expressed as values of *n* (%), mean ± standard deviation (min–max), and median (min–max). Pearson’s chi-square test and Fisher’s exact test were used to analyze the relationships between categorical variables. The Shapiro–Wilk test was used to assess the normality hypothesis. For the analysis of continuous variables, one-way analysis of variance (ANOVA) and independent-samples *t*-tests were employed, with Bonferroni-adjusted post hoc comparisons to control for multiple testing. All statistical analyses were performed using SPSS (Statistical Package for the Social Sciences) v30 (IBM Corp., Armonk, NY, USA). Statistical significance was defined as a two-tailed *p*-value less than 0.05.

## 3. Results

### 3.1. Characteristics of the Participants in the Control, Precancerous Group, and GC Groups

A total of 10 patients were excluded from the final analysis—three due to inadequate biopsy samples, two due to prior gastric surgery history, and five due to the use of medications potentially influencing endocan expression—resulting in the inclusion of 72 patients with precancerous gastric lesions and 80 patients with histologically confirmed gastric cancer. Among these, 27 individuals (37.5%) in the precancerous group and 22 individuals (27.5%) in the GC group were female. The control group was composed of 55 healthy individuals with no known history of gastric pathology, who were matched with the patient groups in terms of age and sex to minimize demographic confounding. The mean ages of patients in the precancerous low-grade dysplasia and high-grade dysplasia subgroups were 56.55 ± 6.30 years and 66.54 ± 10.24 years, respectively (*p* < 0.001). In the GC group, the mean ages were 63.48 ± 10.41 years in stage 1, 63.48 ± 11.87 years in stage 2, and 63.39 ± 11.47 years in stage 3 patients (*p* = 0.955). The mean age (63.39 ± 11.38) in the gastric cancer group was significantly higher than in the control group (58.15 ±11.25) (*p* = 0.019). Statistical analysis revealed no significant differences in age distribution between the precancerous and control groups (*p* = 0.155). According to the TNM staging system, seven patients (8.7%) were categorized as stage 1, 27 (33.7%) as stage 2, and 46 (57.5%) as stage 3 among the gastric cancer patients. These clinical and demographic characteristics are summarized in Table 1, providing a comprehensive overview of the study population’s baseline features.

### 3.2. Endocan Expression in Groups

In the precancerous group, endocan immunohistochemical staining was positive in 47 out of 72 tissue samples, corresponding to 65.3% of the cases. In contrast, all 80 patients in the GC group demonstrated strong endocan positivity. Conversely, none of the healthy control samples (n = 55) exhibited positive staining for endocan, confirming its absence in non-neoplastic gastric mucosa (Table 2). As seen in Table 2, when stratified by sex, a similar pattern emerged: male control samples showed no endocan expression (0/23; 0%), compared to 66.7% positivity in precancerous lesions (30/45) and 100% positivity in cancer specimens (58/58) (*p* < 0.001 for all comparisons). Female tissues likewise transitioned from 0% expression in controls (0/32) to 63.0% in precancerous lesions (17/27) and 100% in carcinomas (22/22) (*p* < 0.001). These findings indicate that endocan upregulation is an early event in gastric tumorigenesis and that its expression correlates strongly with neoplastic progression, independent of patient sex.

Furthermore, when the precancerous group was stratified according to dysplasia grade, a statistically robust association was observed between the severity of epithelial atypia and endocan immuno-expression (χ^2^ = 45.261, *p* < 0.001), as shown in Table 3. Among patients with low-grade dysplasia, the majority—25 out of 33 cases (75.8%)—were negative for endocan, with only a minority (eight cases, 24.2%) showing positive staining. In stark contrast, all 39 cases of high-grade dysplasia exhibited unequivocal positive endocan expression, yielding a 100% positivity rate in this subgroup. Notably, there were no cases of high-grade dysplasia in which endocan staining was absent. When stratified by sex, a concordant pattern emerged: among male subjects, endocan was detected in 28.6% of low-grade dysplasias (6/21) compared with 100% of high-grade dysplasias (24/24; *p* < 0.001), and among females, positivity rates rose from 16.7% in low-grade lesions (2/12) to 100% in high-grade lesions (15/15; *p* < 0.001). These differences were evaluated by chi-square or Fisher’s exact test as appropriate. Collectively, these findings indicate that endocan upregulation is strongly correlated with increasing severity of premalignant gastric mucosal changes, suggesting its potential utility as a biomarker for high-grade dysplasia (Table 3).

## 4. Discussion

In this study, we demonstrated that tissue endocan expression could be a potential new marker for differentiating dysplastic gastric lesions from normal (sensitivity: 60% specificity: 100%) and malignant tissue (sensitivity: 85% specificity: 100%). Moreover, endocan also showed strong performance in grading dysplastic lesions (sensitivity: 100% and specificity: 88%). Given that gastric dysplasia represents the second-to-last phase in the progression of gastric carcinogenesis, it is imperative to focus on the identification, treatment, and surveillance of such lesions. This emphasis on early detection and prevention is crucial for managing gastric cancer [9]. A growing body of evidence suggests that endocan holds promise as a novel marker associated with endothelial cells, contributing to the control of fundamental processes such as cell adhesion, angiogenesis, and the advancement of tumors [17,23].

Endocan is overexpressed in a variety of tumor forms, including glioblastoma, pituitary adenoma, nonsmall cell lung cancer, and colon cancer in recent studies [17]. However, the role of endocan in gastric cancer has been inconsistent in these studies. Zhang et al. found that endocan expression was significantly lower in GC tissues than in normal stomach tissues [24]. In addition, endocan expression and tumor differentiation were found to be inversely related in this study. Therefore endocan is thought to be a tumor suppressor gene, and its downregulation may induce the development of gastric cancers [24]. On the other hand, there exist research findings demonstrating that endocan exhibits overexpression in gastric GC and is linked to distant metastasis, vascular invasion, and tumor staging, as well as diminished overall survival and a lower 5-year survival rate. Consequently, it is suggested that endocan potentially functions as an oncogenic factor in gastric cancer, as it is believed to facilitate the proliferation of cancer cells [19,20,23]. In this study, we found also that, as in many other studies, endocan was overexpressed in dysplastic gastric lesions and gastric cancer. More importantly, this is the first study to show that endocan was also overexpressed in gastric dysplastic lesions, unlike other studies. The biological basis for increased endocan expression in high-grade versus low-grade dysplasia may be linked to its role in promoting epithelial-to-mesenchymal transition (EMT), a key process in early tumor progression. Recent studies have shown that ESM1 overexpression in gastric cancer cells can trigger EMT through activation of the TGF-β and β-catenin signaling pathways [25]. EMT is associated with increased cellular proliferation, motility, and invasiveness—features that may already be primed in high-grade dysplastic lesions. Furthermore, the upregulation of endocan by angiogenic and inflammatory cytokines may reflect a more active pro-tumorigenic microenvironment in high-grade lesions.

Since gastric dysplasia is a neoplastic lesion, and treatment and surveillance of low-and high-grade dysplastic lesions are different, accurate diagnosis and grading are very important in preventing GC. Forceps biopsy is often used for diagnosis, but there are significant diagnostic discrepancies between forceps biopsy and endoscopically resected specimens. Therefore, different endoscopic techniques and methods are being investigated to make the diagnosis more accurate. For instance, Mendoza, P et al. conducted a comparative study between systematic alphanumeric-coded endoscopy and traditional endoscopy to identify precancerous lesions and early gastric cancer. Both methods demonstrated effectiveness, yet systematic alphanumeric-coded endoscopy notably lowered the incidence of false positive results [26]. Chen et al. conducted a study to evaluate the clinical effectiveness of different magnifying chromoendoscopy (MCE) approaches in the early detection of gastric precancerous lesions and cancers. The research demonstrated that MCE methods improve the accuracy of diagnosing early gastric cancer and precancerous lesions [27]. However, in this study, only patients with IM were evaluated as gastric precancerous lesions. In addition, the sample size was small, (six IM and one GC). Recently, the measurement of volatile organic gases in exhaled breath has been used in the diagnosis of many cancers [28]. Amal. H et al. have compared the volatile gas components in the exhaled air of patients with GC or precancerous gastric lesions with a healthy control group [29]. In this study, patients with IM constituted the majority of the gastric precancerous lesion group. The sample size of the patients with dysplasia was quite low (seven gastric dysplasia versus 325 IM). Furthermore, it has been observed that sensitivity and specificity exhibit limitations in distinguishing between low-risk and high-risk lesions. Concurrently, related studies have explored tumor-associated proteins in GC by employing proteome-based methods in both in vitro cell lines and animal models [1,18]. In particular, Li and colleagues conducted an investigation into serum proteomic markers for gastric precancerous lesions and GC, employing matrix-assisted laser desorption/ionization time-of-flight mass spectrometry (MALDI-TOF-MS). Their research indicated that the analysis of serum samples using MALDI-TOF-MS proves valuable in distinguishing between healthy individuals and those with gastric precancerous lesions or GC [1]. However, the sample size of gastric precancerous patients included in this study was small (*n* = 25), and the classification of precancerous lesions was not considered in the study. Moreover, this technique is not cost-effective and provides the *m*/*z* values and not the original peptides or proteins. In the current study, unlike previous research, we have demonstrated that endocan is a robust potential marker for diagnosing and grading gastric dysplastic lesions. Additionally, our study included a significantly higher number of patients with gastric dysplastic lesions compared to other studies. Endocan is a more cost-effective and easily applicable marker than other methods.

This study has several limitations that should be acknowledged when interpreting the results. First, the study was conducted at a single center, which may limit the generalizability of the findings to broader populations or different healthcare settings. Second, the overall sample size—particularly the number of patients with early-stage (T1) gastric cancer—was relatively limited. This was primarily due to the application of strict inclusion and exclusion criteria, which, while intended to minimize confounding variables, may have inadvertently reduced the representativeness of the sample. Third, the study focused exclusively on gastric dysplasia as the sole category of precancerous lesion. Other well-recognized premalignant gastric conditions, such as intestinal metaplasia or gastric adenomas, were not included in the analysis. Although our study focused on tissue-based endocan expression, future investigations may benefit from incorporating noninvasive methods such as serum or plasma endocan measurement using ELISA, which has previously been shown to correlate with tumor progression and prognosis [30]

In conclusion, to the best of our knowledge, this is the first study to provide direct evidence that endocan is overexpressed in gastric dysplastic lesions. Our findings suggest that endocan expression in gastric tissue not only distinguishes dysplastic from non-neoplastic and malignant lesions but also effectively differentiates between low-grade and high-grade dysplasia. Given its high sensitivity, acceptable specificity, and practicality in routine immunohistochemical analysis, endocan emerges as a promising and practical biomarker for both the diagnosis and histopathological grading of gastric dysplasia. These results hold significant potential for enhancing early detection strategies and improving risk stratification in the precancerous phase of gastric carcinogenesis.

## Figures and Tables

**Figure 1 diagnostics-15-01379-f001:**
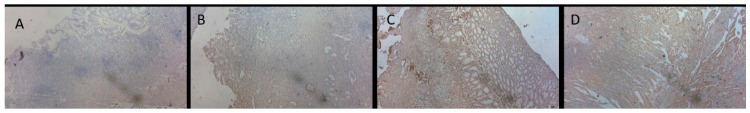
**Immunohistochemical staining of endocan in gastric tissues.** Endocan protein was highly expressed in neoplastic gastric cancer tissues. Immunohistochemical staining of endocan in normal gastric tissues (negative) expression ((**A**), ×40 magnification), 25% (+) expression ((**B**), ×40 magnification), 25–50% (++) expression ((**C**), ×40 magnification), and >50% (++) expression ((**D**), ×40 magnification) in neoplastic gastric tissues. Scale bars represent 50 µm in all panels.

**Table 1 diagnostics-15-01379-t001:** Demographic Characteristics of Participants.

Variable	Subgroup	Group					
		Control	Precancerous Group		Gastric Cancer Group		
			Low Grade	High Grade	Stage-1	Stage-2	Stage-3
Sex, *n* (%)	Male	23 (41.8)	21 (63.6)	24 (61.5)	4 (57.1)	17 (63.0)	37 (80.4)
	Female	32 (58.2)	12 (36.4)	15 (38.5)	3 (52.9)	10 (37.0)	9 (19.6)
Age, *n* (%)	31–40	3 (5.5)	0 (0,0)	0 (0,0)	0 (0,0)	2 (7.4)	1 (2,2)
	41–50	10 (18.2)	3 (9.1)	0 (0,0)	1 (14.3)	1 (3.7)	6 (13.0)
	51–60	19 (34.5)	21 (63.6)	15 (38.5)	2 (28.6)	5 (18.5)	9 (19.6)
	>60	23 (41.8)	9 (27.3)	24 (61.5	4 (57.1)	19 (70.4)	30 (65.2)
Age, x ± SD		58.15 ± 11.25	56.55 ± 6.30	66.54 ± 10.24	63.48 ± 10.41	63.48 ± 11.87	63.39 ± 11.47
Total, *n*		55	33	39	7	27	46

**Table 2 diagnostics-15-01379-t002:** Examination of Participants’ Endocan Staining Status by Group.

Group	Endocan Expression Status		*p*-Value *
	Negative, *n* (%)	Positive, *n* (%)	
Total			
Control group	55 (100)	0 (0)	<0.001
Precancerous group	25 (34.7)	47 (65.3)	
Gastric cancer group	0 (0)	80 (100)	
Male			
Control group	23 (100)	0 (0)	<0.001
Precancerous group	15 (33.3)	30 (66.7)	
Gastric cancer group	0 (0)	58 (100)	
Female			
Control group	32 (100)	0 (0)	<0.001
Precancerous group	10 (37)	17 (63)	
Gastric cancer group	0 (0)	22 (100)	

* Chi-square test.

**Table 3 diagnostics-15-01379-t003:** Examination of Participants’ Endocan Staining Status by Grade.

Premalignant Status	Endocan Expression Status		*p*-Value
	Negative, *n* (%)	Positive, *n* (%)	
Total			
Low grade	25 (75.8)	8 (24.2)	<0.001 *
High grade	0 (0)	39 (100)	
Male			
Low grade	15 (71.4)	6 (28.6)	<0.001 *
High grade	0 (0)	24 (100)	
Female			
Low grade	10 (83.3)	2 (16.7)	<0.001 **
High grade	0 (0)	15 (100)	

* Chi-square test. ** Fisher’s exact test.

## Data Availability

The original contributions presented in this study are included in the article. Further inquiries can be directed to the corresponding author.

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
