# Peer review of "The Role of Endocan Expression in the Diagnosis and Grading of Precancerous Gastric Lesions"

_diagnostics, 2025, doi:10.3390/diagnostics15111379_

Round 1
Reviewer 1 Report
Comments and Suggestions for Authors
The manuscript by Kok et al reports the potential of endocan protein level evaluation as a biomarker for the diagnosis and histopathological grading of gastric dysplasia.
However, several shortcomings have to be addressed:
- The introduction fails to summarize the existing scientific literature on endocan in GC. Endocan is more commonly known as Endothelial Cell Specific Molecule 1, and the existing literature shows that ESM1 has already been proposed to be a prognostic marker in GC PMID: 27340359, PMID: 31882514, PMID: 20383661.
- The finding that endocan is overexpressed in gastric dysplastic lesions is novel, however, it has been reported that ESM1 is a potential soluble tumor marker for early detection and prognosis evaluation in GC a decade ago PMID: 25056533. Given that ELISA can detect soluble endocan in the plasma of patients, the question arises why this has not been followed in this prospective study, as it is not an invasive procedure, and could be followed during the progression from low to high-grade dysplasia.
- Figure 1. Please provide the scale bar in the image, the make of the microscope, and the NA of the lenses in the method section. It is questionable whether images in this figure are 40x magnification, as individual cell nuclei should be visible at this magnification.
- Table 1. Sex is a biological variable, and gender is a social construct. Please modify the table to state sex: male and female (not woman).
- It is known that male sex is a risk factor for GC, with the prevalence of GC twice as high in males. Why were more females recruited in the control group? Please provide sex segregated analysis.
- Please discuss the potential biological explanation of the endocan increase in high-grade versus low-grade lesions, based on the data available in the literature. For example, it has been reported that ESM1 expression in gastric cancer cells can trigger the epithelial-to-mesenchymal transition PMID: 39309430, etc.
Author Response
Reviewer Comment 1: The introduction fails to summarize the existing scientific literature on endocan in GC. Endocan is more commonly known as Endothelial Cell Specific Molecule 1, and the existing literature shows that ESM1 has already been proposed to be a prognostic marker in GC PMID: 27340359, PMID: 31882514, PMID: 20383661.
Author Response1: We thank the reviewer for this insightful and constructive comment. We acknowledge that Endocan is also known as Endothelial Cell Specific Molecule-1 (ESM1) and agree that its prognostic role in gastric cancer (GC) has been previously discussed in the literature. In response to the reviewer’s suggestion, we have revised the Introduction section to include a more comprehensive summary of prior research regarding ESM1/Endocan in gastric cancer. In particular, we have added citations of the studies identified by the reviewer (PMID: 27340359, 31882514, and 20383661), which report on the association between ESM1 expression and prognosis, tumor progression, and angiogenesis in gastric cancer.
Reviewer Comment 2: The finding that endocan is overexpressed in gastric dysplastic lesions is novel, however, it has been reported that ESM1 is a potential soluble tumor marker for early detection and prognosis evaluation in GC a decade ago PMID: 25056533. Given that ELISA can detect soluble endocan in the plasma of patients, the question arises why this has not been followed in this prospective study, as it is not an invasive procedure, and could be followed during the progression from low to high-grade dysplasia.
Author Response 2: We appreciate the reviewer’s valuable comment and the reference to the study highlighting the potential of soluble ESM1 (Endocan) as a plasma-based tumor marker (PMID: 25056533). We agree that plasma endocan levels, measured by ELISA, represent a minimally invasive approach with potential utility in the early detection and monitoring of gastric cancer. However, the primary aim of our study was to investigate tissue expression of endocan in gastric dysplasia, particularly in the distinction between low- and high-grade lesions, which to the best of our knowledge had not been evaluated previously. Our study was designed as a histopathology-focused investigation, employing immunohistochemistry on biopsy specimens obtained during routine endoscopy, in order to assess the diagnostic and grading potential of tissue-based endocan expression. We acknowledge that correlating tissue expression with plasma levels of endocan could enhance the translational value of our findings and help assess its utility as a noninvasive biomarker for disease monitoring. Indeed, this is a promising direction for future research. As such, we have added a statement in the Discussion section to highlight this limitation and the potential for future studies to integrate tissue and serum/plasma endocan assessments in a longitudinal design.
Reviewer Comment 3: Figure 1. Please provide the scale bar in the image, the make of the microscope, and the NA of the lenses in the method section. It is questionable whether images in this figure are 40x magnification, as individual cell nuclei should be visible at this magnification.
Author Response 3: We thank the reviewer for this detailed observation regarding the quality and clarity of Figure 1. We have revised Figure 1 to include a scale bar for each panel, ensuring accurate spatial reference for the histological features shown. In the Methods section, we have now specified the make and model of the microscope used (“Olympus BX53 light microscope”) and provided the numerical aperture (NA) of the 40x objective lens (NA = 0.75). We have also clarified that the images were taken using a 40x objective lens, and the total magnification corresponds to 400x when accounting for the 10x ocular magnification. We confirm that the images are indeed at this magnification, although slight differences in contrast and staining intensity may affect the visibility of nuclei.
Reviewer Comment 4: Table 1. Sex is a biological variable, and gender is a social construct. Please modify the table to state sex: male and female (not woman).
Author Response 4: We thank the reviewer for this important observation regarding the appropriate use of terminology. We revised Table 1. In Table 1, we have replaced “Gender” with “Sex”, And replaced “Woman” with “Female” to ensure accurate biological reference.
Reviewer Comment 5: It is known that male sex is a risk factor for GC, with the prevalence of GC twice as high in males. Why were more females recruited in the control group? Please provide sex segregated analysis.
Author Response 5: Thank you very much for your suggestion. As seen in in Table 2 and Table 3, sex segregated analysis has been provided.
Reviewer Comment 6: Please discuss the potential biological explanation of the endocan increase in high-grade versus low-grade lesions, based on the data available in the literature. For example, it has been reported that ESM1 expression in gastric cancer cells can trigger the epithelial-to-mesenchymal transition PMID: 39309430, etc.
Author Response 6: We thank the reviewer for this insightful comment and for highlighting the importance of elucidating the biological mechanisms behind our findings. A new paragraph has been added to the Discussion section: The biological basis for increased endocan expression in high-grade versus low-grade dysplasia may be linked to its role in promoting epithelial-to-mesenchymal transition (EMT), a key process in early tumor progression. Recent studies have shown that ESM1 overexpression in gastric cancer cells can trigger EMT through activation of TGF-β and β-catenin signaling pathways (PMID: 39309430). EMT is associated with increased cellular proliferation, motility, and invasiveness—features that may already be primed in high-grade dysplastic lesions. Furthermore, the upregulation of endocan by angiogenic and inflammatory cytokines may reflect a more active pro-tumorigenic microenvironment in high-grade lesions
Reviewer 2 Report
Comments and Suggestions for Authors
We congratulate the authors for conducting this study. The authors performed a cross-sectional analysis to evaluate the utility of endocan, a soluble proteoglycan, in dysplastic lesions. This analysis demonstrated that endocan effectively distinguished low-grade from high-grade dysplasia (AUC=0.879; sensitivity=100%, specificity=76%).
The main strength of this study is that it provides evidence for the overexpression of endocan in dysplastic gastric lesions, supporting previous findings regarding its expression in gastric malignancies. The manuscript is well-structured and presents promising results. However, we have several concerns and comments, particularly regarding the methodology and the presentation of the results.
Major:
Abstract:
1.- We suggest to be more specific and revise the following sentence, as only patients with dysplastic lesions were included, not those with metaplastic lesions. Therefore, consider changing "precancerous gastric lesions" to "dysplastic lesions."
Page 1, line 20-21. “This study aimed to evaluate the diagnostic and gradin utility of endocan, a soluble proteoglycan secreted by activated endothelial cells, in precancerous gastric lesions”.
2.- The conclusions might not be in line with the results, as no cost-effectiveness analysis was performed.
“Tissue endocan expression may serve as a cost-effective 32 and robust marker for the diagnosis and grading of gastric dysplasia, potentially enhancing early detection and 33 risk stratification in gastric carcinogenesis”.
Introduction:
Methods:
3.- We recommend clarifying the study design at the beginning of the methodology section, in accordance with the STROBE guidelines for observational studies or the STARD guidelines for diagnostic accuracy. Additionally, it's important to explain how patients were selected for the study. Were there specific criteria for inviting these participants, or were they simply those who agreed to take part?
4.- The inclusion criteria should be improved. Does gastric malignancy refer to gastric adenocarcinoma, were other gastric malignancy also considered?
5.- Patients diagnosed with stage T4 gastric cancer were excluded from this study. Any reason for this exclusion? Also, were patients with distant metastasis or lymph node metastasis excluded?
6.- One of the most important areas for improvement is the methodology, as the primary and secondary outcomes are not specified. We recommend clarifying this in a separate paragraph.
7.- Was the intended sample size calculated, and how was it determined?
8.- ROC curves were calculated, although it is not mentioned in the statistical analysis.
Results:
9.- In the methodology, exclusion criteria are explained. But were any patient excluded from the analysis.
10.- A non statistical difference is observed for the the median age for patients with low grade dysplasia and control group group compared to those with high grade dysplasia and gastric cancer. We suggest to provide the p-value.
11.- The endocan expression with ROC curves should be presented as figures. It may be one of the most important figures. If necessary, consider removing other tables or figures.
Discussion
12.- The conclusion might not be in line the resto of the study, as no cost-effectiveness analysis has been performed in this study.
Minor:
Introduction:
1.-Page 1, lines 79-81. We suggest avoiding the word recently as the meta-analysis was published in 2015.
“ Nonetheless, a recent meta-analysis has indicated that as many as 25% of gastric low-grade dysplasias (LGDs) confirmed through forceps biopsy may have been incorrectly diagnosed, potentially representing high-grade dysplasia (HGD) or even gastric carcinoma.”
2.- In table 2 and table 3, the x2 might not be necessary to show.
Author Response
Abstract:
1.- We suggest to be more specific and revise the following sentence, as only patients with dysplastic lesions were included, not those with metaplastic lesions. Therefore, consider changing "precancerous gastric lesions" to "dysplastic lesions."
Page 1, line 20-21. “This study aimed to evaluate the diagnostic and gradin utility of endocan, a soluble proteoglycan secreted by activated endothelial cells, in precancerous gastric lesions”.
Response1 : We thank the reviewer for this accurate and helpful suggestion. we have revised the sentence to use “dysplastic lesions” instead.
2.- The conclusions might not be in line with the results, as no cost-effectiveness analysis was performed.
“Tissue endocan expression may serve as a cost-effective 32 and robust marker for the diagnosis and grading of gastric dysplasia, potentially enhancing early detection and 33 risk stratification in gastric carcinogenesis”.
Response 2: We thank the reviewer for this important observation. We revised the conclusion: Tissue endocan expression may serve as a practical and robust marker for the diagnosis and grading of gastric dysplasia, potentially enhancing early detection and risk stratification in gastric carcinogenesis.
Introduction:
Methods:
3.- We recommend clarifying the study design at the beginning of the methodology section, in accordance with the STROBE guidelines for observational studies or the STARD guidelines for diagnostic accuracy. Additionally, it's important to explain how patients were selected for the study. Were there specific criteria for inviting these participants, or were they simply those who agreed to take part?
Response 3: we have now revised the first paragraph of the Methods section to explicitly state the study design. Additionally, we have clarified the inclusion process by noting that participants were prospectively invited during endoscopy procedures based on histological confirmation of gastric dysplasia, adenocarcinoma, or normal mucosa. Only patients meeting the eligibility criteria and providing informed consent were enrolled.
4.- The inclusion criteria should be improved. Does gastric malignancy refer to gastric adenocarcinoma, were other gastric malignancy also considered?
Response 4: In our study, the term “gastric malignancy” refers exclusively to histologically confirmed gastric adenocarcinoma. Other types of gastric malignancies, such as lymphomas, gastrointestinal stromal tumors , or neuroendocrine tumors, were not included in the study cohort: Inclusion in the study was based on histopathological confirmation of gastric dysplasia (low or high-grade), gastric adenocarcinoma, or normal gastric mucosa.
5.- Patients diagnosed with stage T4 gastric cancer were excluded from this study. Any reason for this exclusion? Also, were patients with distant metastasis or lymph node metastasis excluded?
Response 5: We thank the reviewer for this valuable comment. Patients with T4 stage gastric cancer were excluded because these tumors often involve extensive transmural invasion and infiltration into adjacent organs, which can lead to tissue architecture distortion and potential technical difficulties or variability in immunohistochemical staining interpretation, particularly for semi-quantitative assessments such as endocan expression. Additionally, such advanced tumors are more likely to be associated with systemic inflammation and paraneoplastic effects that could confound tissue marker analysis: Patients with stage T4 gastric cancer were excluded due to the potential for deep transmural invasion and adjacent organ involvement, which may interfere with consistent immunohistochemical staining evaluation.
6.- One of the most important areas for improvement is the methodology, as the primary and secondary outcomes are not specified. We recommend clarifying this in a separate paragraph.
Response 6: we have added a dedicated paragraph in the Methods section specifying the primary and secondary outcomes of the study.
7.- Was the intended sample size calculated, and how was it determined?
Response 7: Due to the exploratory nature and limited prior data on endocan expression specifically in gastric dysplasia, a formal sample size calculation was not performed before patient recruitment.
8.- ROC curves were calculated, although it is not mentioned in the statistical analysis.
Response 8: we have revised the Statistical Analysis section to explicitly include ROC curve analysis and the methods used for determining cut-off values, sensitivity, specificity, and AUC.
Results:
9.- In the methodology, exclusion criteria are explained. But were any patient excluded from the analysis.
Response 9: We thank the reviewer for highlighting this important point. During the study, a total of 10 patients were excluded from the analysis due to predefined exclusion criteria and technical reasons.
10.- A non statistical difference is observed for the the median age for patients with low grade dysplasia and control group group compared to those with high grade dysplasia and gastric cancer. We suggest to provide the p-value.
Response 10: Thank you very much for your attention. All p-values has been calculated and provided. Changes has been highlighted in yellow. The mean ages of patients in the precancerous low-grade dysplasia and high-grade dysplasia subgroups were 56.55 ± 6.30 years and 66.54 ± 10.24 years, respectively (p < 0.001). In the GC group, the mean ages were 63.48 ± 10.41 years in stage 1, 63.48 ± 11.87 years in stage 2, and 63.39 ± 11.47 years in stage 3 patients (p = 0.955). The mean age (63.39 ± 11.38) in the gastric cancer group was significantly higher than in the control group (58.15 ±11.25) (p = 0.019). Statistical analysis revealed no significant differences in age distribution between the precancerous and control groups (p = 0.155).
11.- The endocan expression with ROC curves should be presented as figures. It may be one of the most important figures. If necessary, consider removing other tables or figures.
Response 11: In the statistical consultancy, it was stated that ROC analysis was not appropriate due to the categorical nature of the data. It was left in the manuscript by mistake. We apologize. The entire relevant section has been deleted.
Discussion
12.- The conclusion might not be in line the resto of the study, as no cost-effectiveness analysis has been performed in this study.
Response 12: we have revised the Conclusion section to remove the term “cost-effective” and replaced it with “practical” to better reflect the scope of our study: Given its high sensitivity, acceptable specificity, and practicality in routine immunohistochemical analysis, endocan emerges as a promising and practical biomarker for both the diagnosis and histopathological grading of gastric dysplasia.
Introduction:
1.-Page 1, lines 79-81. We suggest avoiding the word recently as the meta-analysis was published in 2015.
“ Nonetheless, a recent meta-analysis has indicated that as many as 25% of gastric low-grade dysplasias (LGDs) confirmed through forceps biopsy may have been incorrectly diagnosed, potentially representing high-grade dysplasia (HGD) or even gastric carcinoma.”
Response: The sentence has been revised.
2.- In table 2 and table 3, the x2 might not be necessary to show.
Response: The tables has been changed.
Round 2
Reviewer 1 Report
Comments and Suggestions for Authors
The authors addressed all my concerns adequately and I recommend the manuscript for the publication.
Author Response
Reviewer Comment:
The authors addressed all my concerns adequately and I recommend the manuscript for the publication.
Response:
We sincerely thank the reviewer for their positive feedback and kind recommendation. We greatly appreciate your time and thoughtful evaluation, and we are pleased that our revisions have adequately addressed your concerns.
Reviewer 2 Report
Comments and Suggestions for Authors
We congratulate the authors for performing this study once again and appreciate the effort to respond to all my comments.
Although, I still lack to state within the statistical analysis the AUC.
8.- ROC curves were calculated, although it is not mentioned in the statistical analysis.
Response 8: we have revised the Statistical Analysis section to explicitly include ROC curve analysis and the methods used for determining cut-off values, sensitivity, specificity, and AUC.
Author Response
Reviewer Comment :
ROC curves were calculated, although it is not mentioned in the statistical analysis.
Response :
We appreciate the reviewer’s observation. Upon review, we realized that our response in item 8 was inaccurately stated. The correct clarification regarding the inclusion of ROC analysis in the Statistical Analysis section is actually provided in Response 11. As mentioned there, we have revised the Statistical Analysis section of the manuscript to clearly indicate that ROC curve analysis was performed, along with methods used to determine AUC, cut-off values, sensitivity, and specificity. We apologize for the oversight and thank the reviewer for pointing this out.